# A phase 2 open-label study of the safety and efficacy of weekly dosing of ATL1102 in patients with non-ambulatory Duchenne muscular dystrophy and pharmacology in mdx mice

Ian R. Woodcock[1,2,3]*, George Tachas[4], Nuket Desem[4], Peter J. Houweling[2,3], Michael Kean[5], Jaiman Emmanuel[5], Rachel Kennedy[1,2,6], Kate Carroll[1,2], Katy de Valle[1,2,6], Justine Adams[2], Shireen R. Lamandé[2,3], Chantal Coles[2], Chrystal Tiong[2], Matthew Burton[2], Daniella Villano[1], Peter Button[7], Jean-Yves Hogrel[8], Sarah Catling-Seyffer[1,2], Monique M. Ryan[1,2,3], Martin B. Delatycki[9,10], Eppie M. Yiu[1,2,3]

1 Department of Neurology, The Royal Children's Hospital, Melbourne, Australia, 2 The Murdoch Children's Research Institute, Melbourne, Australia, 3 Department of Paediatrics, University of Melbourne, Melbourne, Australia, 4 Antisense Therapeutics Ltd, Melbourne, Australia, 5 Department of Medical Imaging, The Royal Children's Hospital, Melbourne, Australia, 6 Department of Physiotherapy, University of Melbourne, Melbourne, Australia, 7 McCloud Consulting Group, Sydney, Australia, 8 Institut de Myologie, GH Pitié-Salpêtrière, Paris, France, 9 Victorian Clinical Genetics Service, Melbourne, Australia, 10 Murdoch Children's Research Institute, Bruce Lefroy Centre for Genetic Health Research, Melbourne, Australia

* ian.woodcock@rch.org.au

## Abstract

### Background

ATL1102 is a 2'MOE gapmer antisense oligonucleotide to the CD49d alpha subunit of VLA-4, inhibiting expression of CD49d on lymphocytes, reducing survival, activation and migration to sites of inflammation. Children with DMD have dystrophin deficient muscles susceptible to contraction induced injury, which triggers the immune system, exacerbating muscle damage. CD49d is a biomarker of disease severity in DMD, with increased numbers of high CD49d expressing T cells correlating with more severe and progressive weakess, despite corticosteroid treatment.

### Methods

This Phase 2 open label study assessed the safety, efficacy and pharmacokinetic profile of ATL1102 administered as 25 mg weekly by subcutaneous injection for 24 weeks in 9 non-ambulatory boys with DMD aged 10–18 years. The main objective was to assess safety and tolerability of ATL1102. Secondary objectives included the effect of ATL1102 on lymphocyte numbers in the blood, functional changes in upper limb function as assessed by Performance of Upper Limb test (PUL 2.0) and upper limb strength using MyoGrip and MyoPinch compared to baseline.

**Data Availability Statement:** All relevant data are within the paper and its Supporting information files.

**Funding:** The ATL1102 in DMD clinical trial was funded in its entirety by the sponsor Antisense Therapeutics Ltd. Authors Ms Desem and Dr Tachas are employees of the sponsor and so received payment for services from the sponsor as employees. Ms. Desem and Dr Tachas hold an equity interest in the sponsor. Dr Tachas and Ms Desem along with other sponsor employees and sub-contracted specialists were involved in the study design and data analysis. Authors Dr Woodcock and Dr Ryan were at the time of the trial employees of the Royal Children's Hospital and Murdoch Children's Research Institute and are not affiliated with the sponsor in any way and have not received any direct personal payment or honoraria from the sponsors, nor do they or their family members hold a financial interest or stock in the sponsor company. Dr Woodcock is still an employee of the above institutions, but Dr Ryan has since left the employment to take up public office. Dr Woodcock and Dr Ryan were involved in the trial design as unpaid consultants. As this was a clinical trial, publication was always planned from trial inception. No employees of the sponsor were involved in the data collection, although Ms Desem did liaise closely with the MCRI/RCH site staff and Clinical Trial Organisation throughout the trial. Author Dr Button was paid for services as the study statistician. None of the other authors received any payment from the sponsor to conduct this study. All other authors had input into writing or revising this manuscript.

**Competing interests:** The ATL1102 in DMD clinical trial was funded in its entirety by the commercial sponsor Antisense Therapeutics Ltd. Antisense Therapeutics Ltd is a publicly traded company, listed on the Australian ASX. At the time of the trial, authors Ms Desem and Dr Tachas were employees of the sponsor and so received payment for services from the sponsor as employees. Ms. Desem and Dr Tachas hold an equity interest in the sponsor. Dr Tachas and Ms Desem along with other sponsor employees and sub-contracted specialists were involved in the study design and data analysis. Ms Desem has since left the company and no longer is employed by the sponsor. Authors Dr Woodcock and Dr Ryan were at the time of the trial employees of the Royal Children's Hospital and Murdoch Children's Research Institute and are not affiliated with the sponsor in any way and have not received any direct personal payment or honoraria from the sponsors, nor do they or their family members

## Results

Eight out of nine participants were on a stable dose of corticosteroids. ATL1102 was generally safe and well tolerated. No serious adverse events were reported. There were no participant withdrawals from the study. The most commonly reported adverse events were injection site erythema and skin discoloration. There was no statistically significant change in lymphocyte count from baseline to week 8, 12 or 24 of dosing however, the CD3+CD49d+ T lymphocytes were statistically significantly higher at week 28 compared to week 24, four weeks past the last dose (mean change $0.40 \times 10^9$/L 95%CI 0.05, 0.74; p = 0.030). Functional muscle strength, as measured by the PUL2.0, EK2 and Myoset grip and pinch measures, and MRI fat fraction of the forearm muscles were stable throughout the trial period.

## Conclusion

ATL1102, a novel antisense drug being developed for the treatment of inflammation that exacerbates muscle fibre damage in DMD, appears to be safe and well tolerated in non-ambulant boys with DMD. The apparent stabilisation observed on multiple muscle disease progression parameters assessed over the study duration support the continued development of ATL1102 for the treatment of DMD.

## Trial registration

**Clinical Trial Registration**. Australian New Zealand Clinical Trials Registry Number: ACTRN12618000970246.

## Introduction

Duchenne muscular dystrophy (DMD), a severe, progressive, X-linked genetic muscle disease is the most common muscle disorder in boys, affecting 1 in 5000 live male births worldwide [1]. Boys with DMD have onset of progressive muscle weakness in the first decade of life, with death due to cardiorespiratory failure expected in the late third or early fourth decades [2]. Currently the only disease modifying medical treatment is corticosteroid therapy, which delays loss of ambulation by a median 3 years, to 13 years of age [3–5] but carries a significant treatment burden of adverse effects [5].

DMD is associated with absence of dystrophin from muscle. This causes increased susceptibility to contraction-induced muscle damage, with activation of the innate immune macrophages in turn activating the adaptive immune system T lymphocytes, leading to upregulation of pro-inflammatory cytokines, including the extracellular structural protein osteopontin, resulting in chronic inflammation, fibrosis and reduced muscle strength [6]. CD49d, the alpha chain subunit of integrin very late antigen 4 (VLA-4) is expressed widely on immune cells in this cascade and can bind osteopontin [7]. In patients with DMD, the number of CD49d high expressing T lymphocytes is inversely proportional to ambulation speed, with highest concentration seen in non-ambulant patients [8]. Patients with higher concentrations have more severe weakness and are more likely to lose ambulation before 10 yrs of age despite corticosteroid use, suggesting CD49d may be a biomarker of disease severity or activity [9]. In ex vivo studies a monoclonal antibody to VLA-4 prevented patient T-cell binding to muscle cells and transendothelial migration, highlighting a potential therapeutic avenue [9].

hold a financial interest or stock in the sponsor company. Dr Woodcock is still an employee of the above institutions, but Dr Ryan has since left the employment to take up public office as a Member of the Australian Parliament. Dr Woodcock and Dr Ryan were involved in the trial design as unpaid consultants. Dr Woodcock has received honoraria for work performed including educational activities and attendance at advisory board meetings from pharmaceutical companies Biogen, Novartis, Roche and Avidity and an educational travel bursary to attend an international conference in 2016 from Biogen. Dr Woodcock has received grants for research work from FSHD Global Research Foundation, FSHD Society and Fulcrum Therapeutics. Dr Woodcock has been principal investigator on a number of industry-sponsored clinical trials. None of these disclosures affected the work Dr Woodcock performed on this clinical trial. Dr Ryan has received honoraria for work performed including educational activities and attendance at advisory board meetings from pharmaceutical companies Biogen, Novartis, Roche. Dr Ryan has been principal investigator on a number of industry-sponsored clinical trials. None of these disclosures affected the work Dr Ryan performed on this clinical trial. Dr Yiu has received advisory board honoraria from Biogen and Roche, and has received research support from Biogen, Roche, Pfizer and PTC therapeutics unrelated to the content of this manuscript. Dr Yiu has been principal investigator on a number of industry-sponsored clinical trials. None of these disclosures affected the work Dr Yiu performed on this clinical trial. Prof. Delatycki has received grant awards from NHMRC and is principal investigator in industry sponsored clinical trials including trials sponsored by Rearta and PTC. As this was a clinical trial, publication was always planned from trial inception. No employees of the sponsor were involved in the data collection, although Ms Desem did liaise closely with the MCRI/RCH site staff and Clinical Trial Organisation throughout the trial. As the study statistician, author Dr Button was paid a consultancy fee for his services from the trial sponsor commercial company Antisense Therapeutics Ltd. Authors Dr Houweling, Dr Coles and Dr Tiong were recipients of a grant to perform the MDX studies. This grant was paid by the sponsor Antisense Therapeutics Ltd. None of the other authors received any payment from the sponsor to conduct this study. All other authors had input into writing or revising this manuscript. The authors confirm that the involvement of employees of the sponsor Antisense Therapeutics Ltd in the trial design, data analysis and decision to

ATL1102 is a second-generation immunomodulatory 2'MOE gapmer antisense oligonucleotide which specifically targets human CD49d RNA. After binding to the RNA of CD49d, intracellular RNase H attaches resulting in downregulation of CD49d RNA. ATL1102 has previously been trialled to treat Relapsing Remitting Multiple Sclerosis (RRMS), noting that CD49d high expressing T cells are the effector and central memory T cells in RRMS. In this phase 2 RRMS clinical trial, ATL1102 dosed 200mg three times weekly in the first week and twice weekly to 8 weeks substantially reduced inflammatory brain lesions by 88.5% and circulating lymphocytes and T lymphocytes by 25% [10].

Reported here, collaborators from the same institution ran two separate but complementary studies. The initial trial, a phase 2 clinical trial examining for the first time the safety and efficacy of a low dose ATL1102 treatment for 24 weeks in non-ambulant patients with DMD on concomitant corticosteroid treatment. The second, a pre-clinical study in the *mdx* mouse model for DMD, conducted using a mouse specific second generation CD49d ASO (ISIS 348574) to show that monotherapy treatment can reduce CD49d mRNA expression in muscle and decrease contraction induced muscle damage.

## Methods

### Ethics statement

The clinical trial received approval from the Royal Children's Hospital Human Research Ethics Committee with assigned number HREC/17/RCHM/121. The trial was subsequently registered at the Australian New Zealand Clinical Trials Registry (ACTRN12618000970246). An independent Data Safety Monitoring Board (DSMB) was established to provide safety oversight for the trial. Participant consent to participate in the trial was sort from parents. Pre-clinical *mdx* mouse analyses were approved by the Murdoch Children's Research Institute (MCRI) animal care and ethics committee (ACEC; approval number A899).

### Pre-clinical studies in mice

The *mdx* mouse model is commonly used to study DMD. We tested efficacy of the second generation 2'MOE gapmer mouse specific CD49d ASO ISIS 348574 as ATL1102 is specific to human CD49d RNA and not homologous to mouse. Mdx mice do not have circulating lymphocytes with high CD49d but have high CD49d expressing lymphocytes in the lymph nodes at 9 weeks [11]. Symptomatic 9 week old *mdx* mice were treated for 6 weeks to determine the effect of ISIS 348574 on *CD49d* mRNA expression and muscle function measures.

Male *mdx* and age matched C57Bl10/J wild-type controls were purchased from the Jackson laboratories at 5 weeks of age and acclimatised to the MCRI facility for a total of 4 weeks. Animals were housed in a specific-pathogen-free environment at a constant ambient temperature of 22°C and 50% humidity on a 12 h light-dark cycle, with *ad libitum* access to food and water. The *mdx* mice (n = 12 /group) were randomly assigned to 4 treatment groups (saline, low (5mg/kg) and high (20mg/kg) dose ISIS 348574, and a control gapmer oligonucleotide with the same 20 nucleotides scrambled such that it is not complementary to CD49d or other RNA (20mg/kg)). *Mdx* Mice received weekly subcutaneous injections of either saline, ISIS348574 and control oligonucleotide for a total of 6 weeks. Wild-type controls (n = 12) received saline only. The high dose (20mg/kg/week) equates to 1.6mg/kg/w dose in human equivalent body surface area (BSA) and the low dose (5mg/kg/week) equates to 0.4mg/kg/week dose on BSA. After treatment mice were anaesthetised using inhaled isoflurane (0.6 ml per min) and muscle function was examined using the Arora Scientific 1300A whole mouse test system and 701C stimulator as previously published [12]. Mice were then euthanised by cervical dislocation and the spleen and skeletal muscle (quadriceps) were collected for further analysis.

publish this data does not alter our adherence to PLOS ONE policies on sharing data and materials.

## Flow cytometry

The *mdx* spleen was perforated to isolate the splenic sub-cellular content and incubated in red blood cell lysis buffer (Thermofisher) for ten minutes at 4˚C. Splenocytes were centrifuged 1500 g for five minutes at 4˚C. Cell pellets were washed in wash buffer (PBS:1% BSA (bovine albumin serum)). CD4+ and CD8+ T cell populations were identified using anti-CD4-V450 (Biolegend, San Diego, CA, USA) and anti-CD8a-allophycocyanin–cyanine 7 (anti-CD8a-APC-Cy7) (Biolegend, San Diego, CA, USA). Stained cells were analysed using BD LSRFortessa™ X-20 Cell Analyzer to identify populations of CD4+ and CD8+ T cells.

## Clinical trial design

This was a phase 2 open-label clinical trial assessing the safety of ATL1102 in non-ambulant boys with DMD concomitantly with their usual corticosteroid therapy, in all but one participant who discontinued corticosteroids years prior to the study. CONSORT flow diagram of study design is shown in Fig 1. Eligibility criteria for the clinical trial are included as S1 Fig. Participants were recruited from a single site in Melbourne, Australia with the study running from August 2018 until January 2020.

After providing written informed consent, participants received weekly subcutaneous injections of 25mg of ATL1102 for twenty-four weeks. Injections were administered into subcutaneous fat of the abdomen by a registered nurse or trained parent. Injection sites were rotated in quadrants around the umbilicus. Parents monitored injection sites for cutaneous reactions and participant discomfort for forty-eight hours post-administration. Adverse events were recorded in a diary which was returned to the study coordinators at each fortnightly visit.

Participants underwent fortnightly venepuncture for exploratory and safety blood tests. This included monitoring of haematology, biochemistry and inflammatory markers and at point of care urinalysis dipstick to monitor kidney function. Participants were seen monthly by the study team for physical examination and respiratory function assessments.

Participant safety was monitored routinely and at regularly scheduled meetings by the independent DSMB.

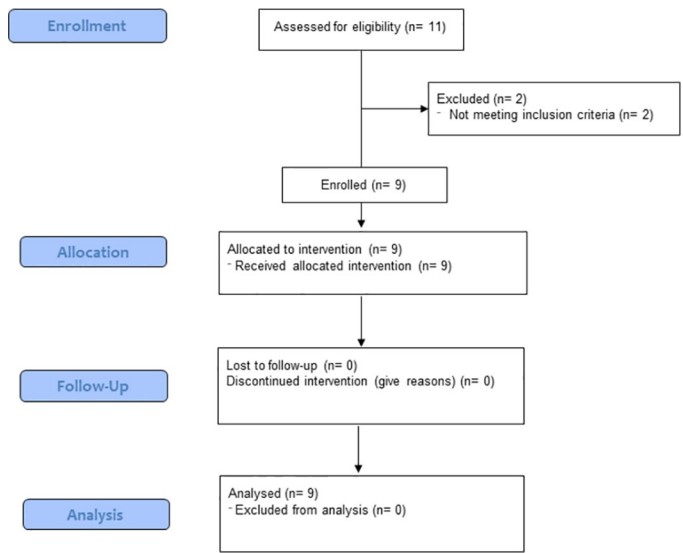

**Fig 1. CONSORT flow diagram of study design.**

## Outcome measurements

The primary endpoint of the trial was safety of ATL1102 as assessed by the frequency and intensity of adverse events, including injection site reactions and any laboratory value derangement.

Secondary outcome measures included both laboratory and functional efficacy endpoints. Laboratory efficacy outcome measures included lymphocyte-modulation activity determined by cell surface flow cytometry measuring variation in the number and percentage of total lymphocytes as well as those CD4 and CD8 T lymphocytes expressing high levels of CD49d (CD49dhi) to week twenty-four of treatment and to week twenty-eight, four weeks past the last treatment.

Changes in upper limb muscle strength and function were measured at baseline and again at weeks five, eight, twelve and twenty-four. Muscle function was assessed by a questionnaire-based outcome measure of disease burden (Egen Klassifikation Scale version 2—EK2) and performance-based measures (the Performance of the Upper Limb scale version 2 (PUL 2.0), and the Moviplate 30 second finger tapping score of the MyoSet tool). The Myoset tool also measured distal upper limb strength as determined by the MyoPinch (key pinch strength) and MyoGrip (hand grip strength) scores [13–15]. Other outcome measures assessed were respiratory function (forced vital capacity (FVC) and forced expiratory volume in one second (FEV1)), and quality of life assessed using the neuromuscular module of the Pediatric Quality of Life Instrument (PedsQL NMD™).

## Muscle Magnetic Resonance Imaging (MRI)

Participants underwent MRI of the dominant forearm at baseline, week twelve and week twenty-four. Unilateral upper-limb MRI was performed at 1.5T (Siemens Aera; Siemens, Erlangen, Germany) using a flexible surface matrix coil (4-Channel Flex Coil) wrapped around the forearm. Participants lay in the scanner in the head-first supine position, with the arm to be imaged lying in a comfortable position on the scanner bed alongside the torso. Two point-Dixon images were acquired (3D gradient-echo TE1/TE2/TR = 2.39/4.44/6.99ms, flip angle 10˚, nine 6mm axial slices, slice gap 0mm, FOV 18x18cm, matrix 320×320, pixel size 0.56×0.56mm, NEX = 4). Fat fraction maps were obtained using on scanner tools.

Change over time in muscle composition (atrophy, oedema and fatty infiltration) was measured on the muscles of the central, proximal, and distal forearm using Short Tau Inversion Recovery (STIR) and 3-point Dixon sequences on MRI. Changes in muscle composition were scored using the semi-quantitative visual scoring Mercuri method and by quantitative fat fraction analysis [16–18]. Due to fatty infiltration, identification of individual muscles was challenging, such that a compartment composite score of volar, dorsal and ECRLB Br (extensor carpi radialis longus/brevis and brachioradialis) compartments was used as per a previous published study [19]. The lean muscle mass was calculated using previously published methods: Cross-sectional muscle compartment area x ((100 –total muscle compartment fat percent) / 100) [19].

## Statistical analysis

Based upon data from a previous study of ATL1102 in RRMS patients analyzing blood 3 days after the last dose in week 8, the laboratory efficacy end point of lymphocyte modulation potential was established as a reduction in total lymphocyte count of $0.47x10^9$/L (25% reduction) [10]. For the sample size calculation, the level of significance was set to 0.05 with a 2-sided paired t-test, mean difference of 0.47 ($x10^9$/L) from baseline to end of treatment, and standard deviation of 0.428 ($x10^9$/L). Using nQuery (Version 8.5.2.0, Table MOT1-1 Paired t-

test for differences in means, Statistical Solutions Ltd.), a sample size of 9 participants was calculated as required to achieve a power of 80%. Nine participants were considered sufficient to investigate the safety, tolerability and PK and PD profile of ATL1102 in this rare target patient population. Data were analysed using SAS® Version 9.4. primary and secondary efficacy measures were analysed using the paired t-test and the non-parametric Wilcoxon sign-rank test. The study was not powered to see a change on the secondary efficacy endpoints from baseline to end of treatment. The study protocol and statistical analysis plan are available as supplementary data in "Protocol" and eSAP1 and eSAP2 respectively. Repeated measures analysis of lymphocytes and T lymphocyte and NK lymphocyte subsets in a *post hoc* analysis was conducted comparing baseline to a linear combination of values measured three days post-dose at weeks 8, 12 and 24. Associations between variables was tested using a Pearson correlation test.

Preclinical *mdx* mouse analyses were performed in Graphpad Prism (V9, Graphpad Software Inc.). For *in situ* muscle function, one-way ANOVA with Tukey correction for multiple testing was performed (n = 9 animals / treatment). Unpaired T-Tests were used for CD4 and CD8 T-cell analyses (n = 5–6 samples / treatment). All data shown as mean with 95% confidence intervals, unless otherwise stated in the figure legends.

## Results

### Proof of concept preclinical study using mdx mice

*Ex vivo* analyses of monocytes collected from *mdx* mice (n = 3), showed that the ASO to mouse CD49d, ISIS 348574 can reduce the expression of CD49d mRNA (Fig 2A). We then examined the *in vivo* response to ISIS 348574 in *mdx* mice treated for 6 weeks which showed that *CD49d* mRNA expression was reduced in skeletal muscle by approximately 40% when treated with either a low (5mg/kg/week) or high (20mg/kg/week) dose of ISIS 348574, compared to saline controls (Fig 2B, One-way ANOVA, summary $p<0.01$, with Tukey correction displayed on the graphs, p = * $<0.05$, ** $<0.01$).

This study also found that *mdx* mice treated with the 20mg/kg/week dose of ISIS 348574 showed a reduction in the percentage of splenic CD4+ (30%, $p<0.05$) and CD8+ (21%, p = 0.058) T lymphocytes, compared to saline treated mice (Fig 2C and 2D). Furthermore *in situ* muscle funcation analyses show that the high dose (20mg/kg/week) treated *mdx* mice were protected from the effects of eccentric muscle damage, producing 72% of the original muscle force ($P<0.01$). This was in contrast to *mdx* mice treated with either saline, scrambled control or low dose (5mg/kg) ISIS 348574, which generated approximately 50% of the original force following eccentric muscle contractions. (Fig 2E and 2F, One-way ANOVA summary p = $<0.001$, with Tukey correction displayed on the graphs, p = * $<0.05$, ** $<0.01$, *** $<0.001$).

### Clinical trial results

Eleven adolescent males with DMD were screened for participation. All had been non-ambulant for at least six months prior to screening. Two screened participants were excluded (one participant had started cardioprotective medication within three months of the initial visit and the other participant exceeded the pre-determined weight limit for inclusion).

Nine participants were enrolled into the open-label study. All had a confirmed pathogenic variant in DMD, with a clinical phenotype consistent with DMD as assessed by their treating/ referring clinician and the study investigator. Participant demographics are summarized in Table 1.

**Safety.** There were no serious adverse events (SAEs) or suspected or unexpected serious adverse reactions (SUSARs). A total of 136 adverse events were recorded (Table 2), with all

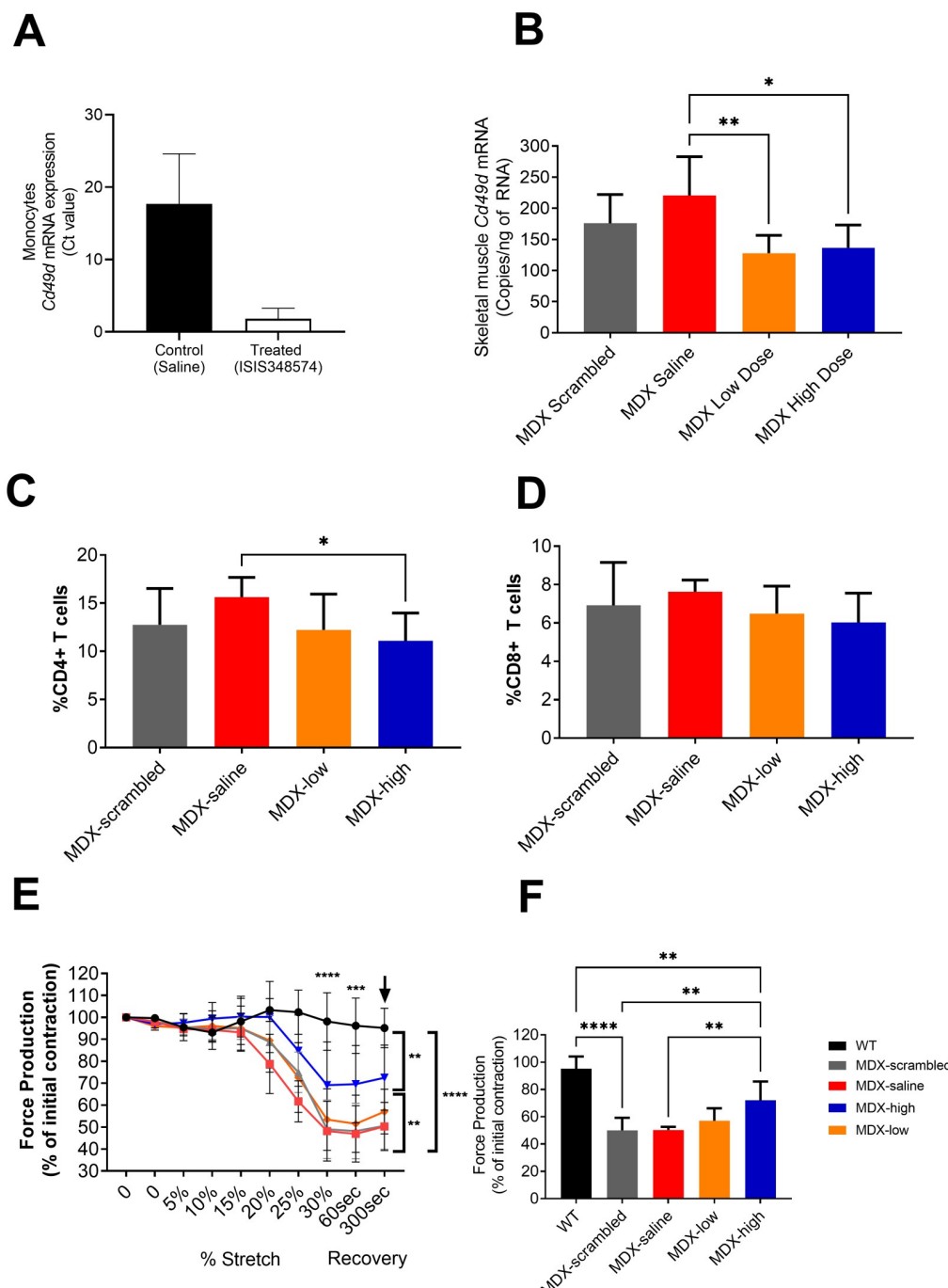

**Fig 2. Pre-clinical data using the mdx mouse model of DMD to test the effects of ISIS 348574 (mouse specific Cd49d oligonucleotide to ATL 1102) in vivo.** A) Monocytes were isolated from the spleens of mdx mice (n = 3) and exposed to a single dose of ISIS 348574 for 48hrs in vitro to show that we could achieve a reduction in CD49d mRNA using ISIS 348574 to mouse CD49d RNA. B) Following 6 weeks of treatment CD49d mRNA expression was reduced in mice treated with either the low (5mg/kg/week) or high (20mg/kg/week) dose of ISIS 348574, compared to saline controls (One-way ANOVA with Tukey correction, p = * <0.05, ** <0.01). C and D) Proportion of CD4+ and CD8+ T cells from the spleens of mdx mice with and without ISIS 348574 drug treatment. Cells are expressed as a proportion of total live cells isolated from the spleen. One way ANOVA with Fishers LSD test, * p < 0.05. E and F) In situ muscle physiology analyses shows that mdx mice treated with either saline (red, ~45% force recovery), scrambled (grey, ~45% force recovery) or low dose (orange, ~ 50% force recovery) ISIS 348574 were susceptible to eccentric muscle contraction damage compared to wild-type (black) controls, whereas the mice treated with a high dose of ISIS 348574 were resistant to the effects of eccentric muscle damage and produced 72% of the original force following the eccentric

muscle damaging protocol. This was still significantly less than the 95% force recovery seen in WT mice, however this improvement in force following a muscle damage protocol suggests that the use of a 20mg/kg/week dose of ISIS 348574 was able to protect the muscles of mdx mice. One-way ANOVA with Fishers LSD test, p = * <0.05, ** <0.01, *** <0.001), ****<0.0001).

participants reporting at least one adverse event. Sixty-three percent of the reported adverse events were injection site related, with all but one participant experiencing transient erythema within twenty-four hours of the administration of ATL1102. Six (67%) participants had mild post-inflammatory hyperpigmentation of the skin of their abdomen which was persistent; four had resolved and two were ongoing but improving at the post completion follow-up study visit. The hyperpigmentation was noticed in the first participant after receiving eleven weekly doses of ATL1102. The DSMB was made aware and the participant informed consent form updated. In the five subsequent participants who had a similar reaction it was seen after four to eleven doses. The hyperpigmentation was not regarded as a clinical safety concern by the DSMB. Pain, discomfort or atrophy of the subcutaneous tissues were not reported, and there were no signs of systemic involvement. No participants withdrew from the study. There were no other significant adverse events felt to be related to ATL1102 or its administration.

**Efficacy.** *Lymphocyte Count*: There was no statistically significant decrease in lymphocyte count from baseline to week eight, week twelve or week twenty-four of dosing (Table 3). There was no statistically significant decrease in CD49d+CD3+CD8$^+$ or CD49d+CD3+CD4$^+$ T lymphocytes seen between baseline, weeks 8, 12 or 24 (Table 3). This 9 participant trial did not achieve the pre-specified laboratory activity outcome measure of a significant -0.47x10$^9$/L (25% reduction) in total lymphocyte count.

There was, though, a consistent trend toward declines in the mean number of lymphocytes, and CD49d+ T lymphocytes measured 3 days post-dose at week 8, 12 and 24. The mean number of CD3+CD49d+ T lymphocytes (i.e. CD3+CD4+ and CD3+CD8+ expressing CD49d) measured at week 28 was statistically significantly higher compared to end of dosing at week 24 (mean change 0.40x10$^9$/L 95%CI 0.05, 0.74; paired T-Test, p = 0.030) (Table 3). Repeated measures analysis of CD3-CD49d+ NK lymphocytes in a *post hoc* analysis comparing baseline to a linear combination of values measured three days post-dose at weeks 8, 12 and 24 was significantly lower compared to baseline (p = 0.018), with comparable NK lymphocyte numbers at week 28 (Fig 3).

**Functional outcome measures.** There were no statistically significant changes in any upper limb functional outcome measures at week 24 compared to baseline (Table 4). The PUL2.0 score remained stable with no significant change between baseline and week twenty-

**Table 1. Summary of participant demographics.**

| Characteristic | Category | Statistic | ATL1102 N = 9 |
|---|---|---|---|
| Sex | Male | n (%) | 9 (100) |
| Age (years) | | Mean (SD) Median (range) | 14.9 (2.1) 14.0 (12–18) |
| Weight (kg) | | Mean (SD) | 52.7 (9.8) |
| Height (cm) | | Mean (SD) | 141.1 (10.0) |
| BMI | | Mean (SD) | 27.1 (7.4) |
| Time since non-ambulant (years) | | Median (range) | 2.2 (0.6–9.2) |
| Corticosteroid Medication | Yes *Prednisolone* *Deflazacort* | n (%) | 8 (88.9) *3 (33.3)* *5 (55.6)* |

**Table 2. Treatment emergent adverse events reported in at least two participants.**

| SYSTEM ORGAN CLASS<br>Preferred Term | All<br>N = 9<br>Participants (%) [No. of Events] |
|---|---|
| Participants reporting any AEs | 9 (100.0%) [136] |
| *GENERAL DISORDERS AND ADMINISTRATION SITE CONDITIONS* | |
| Injection site erythema | 8 (88.9%) [59] |
| Injection site pain | 5 (55.6%) [7] |
| Injection site swelling | 3 (33.3%) [6] |
| Injection site bruising | 4 (44.4%) [4] |
| Pyrexia | 2 (22.2%) [4] |
| *SKIN AND SUBCUTANEOUS TISSUE DISORDERS* | |
| Skin discolouration | 6 (66.7%) [7] |
| *GASTROINTESTINAL DISORDERS* | |
| Vomiting | 2 (22.2%) [4] |
| Constipation | 2 (22.2%) [2] |
| *RESPIRATORY, THORACIC AND MEDIASTINAL DISORDERS* | |
| Cough | 2 (22.2%) [2] |
| Nasal congestion | 2 (22.2%) [2] |
| Oropharyngeal pain | 2 (22.2%) [2] |
| *INFECTIONS AND INFESTATIONS* | |
| Lower respiratory tract infection | 2 (22.2%) [2] |
| Nasopharyngitis | 2 (22.2%) [2] |
| *NERVOUS SYSTEM DISORDERS* | |
| Migraine | 2 (22.2%) [2] |

four with a mean increase in PUL2.0 score of 0.9 (95%CI -1.33, 3.11, where a higher score indicates better function) EK2 scores were stable throughout the trial period. ATL1102 treatment showed no significant effect on lung function throughout the 24 week trial period (Table 4). The components of the Myoset all reflected stable grip and pinch strength over the course of the trial.

**Table 3. Summary of lymphocytes mean change from baseline to weeks 24 and 28.**

| White blood cell type (X10^9 cells per litre) | Mean Baseline count (x10^9 cells per litre) | Mean Change from baseline | | | | Median percentage change from baseline (%) | | | | Paired T-Test (p value) of mean change |
|---|---|---|---|---|---|---|---|---|---|---|
| | | Week 8 | Week 12 | Week 24 | Week 28 | Week 8 | Week 12 | Week 24 | Week 28 | between week 28 and 24 |
| **Lymphocytes** | 3.68 | -0.56 | -0.53 | -0.28 | +0.19 | -4.63 | -7.14 | -4.22 | +11.81 | 0.051 |
| **CD3+ T cells** | 2.93 | -0.53 | -0.33 | -0.18 | +0.25 | -10.9 | -5.46 | 0.86 | +17.11 | 0.056 |
| **CD3+ CD49d+ T cells** | 2.44 | -0.50 | -0.39 | -0.28 | +0.11 | -12.3 | -10.0 | -9.78 | +9.93 | 0.03* |
| **CD4+ T cells** | 1.57 | -0.30 | -0.20 | -0.15 | +0.11 | -12.3 | -5.23 | -1.12 | +16.50 | 0.063 |
| **CD4+ CD49d+ T cells** | 1.20 | -0.28 | -0.22 | -0.19 | +0.01 | -12.5 | -7.09 | -16.7 | +1.73 | 0.073 |
| **CD8+ T cells** | 1.22 | -0.20 | -0.09 | -0.02 | +0.14 | -9.35 | -5.21 | -2.62 | +17.99 | 0.068 |
| **CD8+ CD49d+T cells** | 1.17 | -0.22 | -0.12 | -0.05 | +0.11 | -10.9 | -7.32 | -5.79 | +13.37 | 0.064 |

The Lymphocyte mean number of cells at week 24 (at the end of dosing) is trending significantly lower vs week 28 (p = 0.051 paired T test).

*The mean number of CD3+CD49d+T lymphocytes (CD4+CD49d+ and CD8+CD49d+ T lymphocytes) at week 24 is statistically significantly lower vs week 28 (p = 0.030 paired T test).

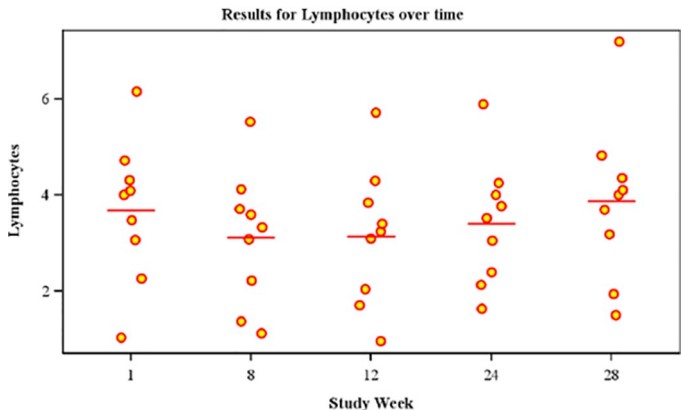

**Fig 3. Showing lymphocyte baseline, week 8, 12 and 24 week data 3 days post each dose and the week 28 data, four week past end of dosing data is shown as a bee swarm plot with mean expression values of lymphocytes.**

**MRI.**  In this trial, there was some variation in the quality of proximal and distal slices due to variable positioning of the participant's forearm in consecutive scans. The quality of the central slices through the muscle body in the forearms muscles was not compromised and so this measurement was chosen for detailed comparison between baseline and week 24 scans for each participant. No significant change was apparent between baseline and week twenty-four for the mean Mercuri semi-quantitative score of fatty infiltration (0.1 point change), atrophy (0 point change) and muscle oedema (0.3 point change) measuring the central slice around the elbow (Table 5). There was a trend towards minor improvement in the percentage fat fraction in all muscle groups measuring the central slice, although statistical significance was not achieved. There was no significant pattern of change in cross-sectional muscle area of any muscle group (Table 6).

**Table 4. Change in functional outcome measures from baseline to week 24.**

| Patient No. | PUL 2.0 | MyoGrip (dom) (Kg) | MyoGrip (dom) (% Pred) | MyoPinch (dom) (Kg) | MyoPinch (dom) (% Pred) | MoviPlate Score (dom) | % Predicted FVC | % Predicted PEF | EK2* |
|---|---|---|---|---|---|---|---|---|---|
| | | | | | Change from Baseline to Week 24 | | | | |
| 01–001 | +2 | -0.63 | -4.49 | 0.03 | -0.62 | -14.0 | -3.20 | 6.30 | +1 |
| 01–002 | +2 | 0.22 | 0.49 | -0.02 | -0.29 | 13.0 | -14.8 | -17.3 | +1 |
| 01–003 | 0 | 0.68 | 1.02 | -0.40 | -6.59 | -3.0 | -9.10 | 8.70 | +2 |
| 01–004 | +2 | 1.09 | 1.01 | 0.37 | 2.99 | 7.0 | 0.80 | 7.20 | +2 |
| 01–006 | -3 | -0.27 | -0.60 | 0.07 | 0.94 | 8.0 | -6.50 | 6.90 | -6 |
| 01–008 | +7 | 1.00 | 1.11 | 0.30 | 2.77 | 3.0 | -7.70 | -18.2 | -1 |
| 01–009 | 0 | -0.33 | -3.75 | -0.22 | -4.97 | 7.0 | -9.10 | -4.30 | +2 |
| 01–010 | 0 | 0.05 | 0.11 | 0.06 | 0.72 | -15.0 | -0.40 | 9.20 | -1 |
| 01–011 | -2 | 0.11 | -1.31 | -0.18 | -3.63 | 11.0 | -1.10 | 2.00 | +2 |
| | | | | | | | | | |
| Mean Change (95% CI): | +0.9 (-1.33, 3.11) | +0.2 (-0.25, 0.67) | -0.7 (-2.33, 0.90) | 0.0 (-0.18, 0.19) | -1.0 (-3.56, 1.63) | 1.9 (-6.08, 9.85) | -5.68 (-9.60, -1.76) | 0.06 (-8.33, 8.44) | 0.2 (-1.80, 2.25) |

*Higher score = greater disability.

#Reduction in Fat Fraction (%) = improvement.

**Table 5. Mercuri visual semi-quantitative score in whole forearm compartment from baseline to week 24.**

| | Whole Forearm | | |
|---|---|---|---|
| | Baseline | Wk 24 | Change |
| *Fatty Infiltration* | 5.2 | 5.3 | 0.1 |
| *Atrophy* | 1.6 | 1.6 | 0.0 |
| *Oedema* | 2.6 | 2.9 | 0.3 |

**Table 6. Change in the MRI Central reading fat fraction, cross sectional area and lean muscle mass from baseline to week 24.**

| Mean Change (95% CI) from Screening/Baseline to Week 24 | | |
|---|---|---|
| MRI Parameter | N | MRI Central Reading Mean (95%CI) |
| *Fat Fraction (%)* | | |
| Volar Muscle | 9 | -0.57 (-7.81, 6.68) |
| Dorsal Muscles | 9 | -0.88 (-3.41, 1.65) |
| ECRLB-Br | 9 | -0.12 (-6.42, 6.17) |
| **Average Fat Fraction** | 9 | -0.52 (-5.62, 4.58) |
| | | |
| *Cross Sectional Muscle Area (mm$^2$)* | | |
| Volar Muscle | 9 | 22.78 (-31.2,76.73) |
| Dorsal Muscles | 9 | 0.89 (-18.9,20.65) |
| ECRLB-Br | 9 | -1.33 (-8.94, 6.28) |
| **Total Area** | 9 | 22.33 (-36.8,81.42) |
| **Lean Muscle Mass** *(mm$^2$)* | 9 | 13.9 (72.6, 100.4) |

ECRLB-Br = extensor carpi radialis longus/brevis and brachioradialis. Volar Muscles; flexor digitorum profundus and flexor pollicis longus (FDP), flexor digitorum superficialis and palmaris longus (FDS), flexor carpi ulnaris (FCU), flexor carpi radialis (FCR). Dorsal Muscles: Extensor carpi ulnaris (ECU), extensor digiti minimi (EDM), extensor digitorum (ED), extensor pollicis longus (EPL), abductor pollicis longus (APL), extensor carpi radialis longus/brevis and brachioradialis (ECRLB-Br), but the ECRL-BR are not included in the Dorsal muscle measurement in the Central Reading. Change in the MRI Proximal reading average Fat Fraction (%) from baseline to week 24 was -2.14 [95%CI -7.60; 3.3] for the 9 patients.

**Correlation of parameters assessed in the Phase 2 study.** Correlation analyses were performed across assessment measures PUL2.0, Myoset and MRI.

Positive correlations were observed in the Phase 2 study between the different measures of muscle function of Moviplate scores and the PUL 2.0 scores of the distal domain ($r = 0.664$) which support the consistency of the observed changes across the measures assessed in the study over the 24 week ATL1102 treatment period (S2 Fig).

Positive correlations were also observed in the Phase 2 study between the MRI results of the lean muscle area (non-fat) and MyoGrip results ($r = 0.604$), suggesting a consistency of results across the different parameters of muscle structure and muscle strength (S3 Fig).

## Discussion

### Safety

This open-label phase 2 clinical trial met its primary safety end point. All but one participant experienced post-injection site erythema, swelling or discomfort suggesting that the investigation product is a mild irritant, as has been observed with other MOE antisense drugs, and

is commonly reported with subcutaneiously injected drugs. Future clinical trials of ATL1102 could consider including using ice as a pre-injection site treatment to minimise these reactions. Six participants experienced an unexpected post-inflammatory skin hyperpigmentation which resolved or faded at the completion of the study. Interestingly this reaction has not been previously reported in other clinical trials of ATL1102 and was not viewed as a safety concern [10].

The dose chosen for this 24 week trial (25mg/week) was considered as a presumed safe dose in this patient population. In a previous phase 2 trial in individuals with RRMS a loading dose of 200mg every other day for one week was administered then 400mg/week (twice weekly 200mg) for seven weeks [10]. This DMD trial was the first clinical trial to investigate the safety of ATL1102 over a six-month period.

## Lymphocyte modulation

Given the previously observed action of ATL1102 of reducing lymphocytes in the RRMS study, the trial sample size was calculated to see a 25% reduction in total lymphocyte count when the drug is at equilibrium from week 8. This activity endpoint was not met in this trial. There was however a consistent trend toward declines in the mean number of lymphocytes at week 8, 12 and 24 each measured 3 days past dosing, and statistically significant reductions in the mean number of CD49d+ NK lymphocytes at week 8, 12 and 24 weeks of treatment, using repeated measures analysis. The mean number of CD49d+ T lymphocytes (i.e CD3+CD4 + and CD3+CD8+ that are CD49d+) was statistically significantly higher at week 28 compared to week 24, indicating a rebound elevation of CD49d+ T lymphocytes four weeks post the last treatment dose. These results collectively suggest that ATL1102 suppresses CD49d expressing lymphocytes at a dose of 25mg per week. It is anticipated that higher doses will increase the level of lymphocyte reduction whilst maintaining a favourable safety profile in part due to sparing of the majority of T lymphocytes and NK lymphocytes. Future studies will look at dose escalation as supported by this study, and modelling with ATL1102.

## PUL2.0 and EK2 upper limb function

PUL2.0 measures shoulder, elbow, and wrist finger dimensions of disease burden and is a reliable measure of disease severity and progression in DMD where a lower score indicates loss of function. The mean increase from baseline to week twenty-four in the PUL2.0 was 0.9 (95% CI -1.33 to 3.11). Although the Minimal Clinical Important Difference (MCID) for the PUL2.0 has not been established, an external historical cohort with same inclusion criteria as in the ATL1102 phase 2 trial, showed a decrease in PUL2.0 score of 2.0 (standard deviation 3.02) from baseline over a six month period [20]. In the ATL1102 phase 2 study four of the nine patients achieved an increase in their PUL2.0 score of +2, and another three patients were stabilized in the PUL2.0 score (Table 4). This is an encouraging trend that warrants further investigation. A previously published data from a historical cohort also reported that over a twelve month period a mean decrease in PUL2.0 score of 2.17 can occur, albeit in a cohort not directly comparable to the participants in the phase 2 study due to older age and larger proportion not on corticosteroids [21].

There was no change in the EK2 over the course of the trial period. This composite outcome measure encompasses multiple aspects of disease burden and as such is a useful clinical monitoring tool (higher score equals greater disease burden) but is not likely to be as responsive as the PUL2.0 measure to small changes in upper limb function. As such, a stabilisation over the six month trial periods is encouraging and needs to be confirmed in future studies.

### Myoset tests: MoviPlate, MyoGrip, and MyoPinch

The MyoSet functional outcome measures consist of the MoviPlate muscle function assessment of repetitive flexion extension of the wrist and fingers, and MyoGrip and MyoPinch assessment of muscle grip and pinch strength [13,14]. These measures have been validated for use in clinical trials of DMD; MyoGrip and MyoPinch in particular have been shown to be sensitive to change in non-ambulant boys and to correlate well with lean muscle mass on MRI [15]. There was no change in any of these measures over the trial period. Previous natural history studies have shown significant deterioration over six months (Grip -0.5kg [95%CI -1.01; 0.002] and Pinch -0.38kg [95%CI -0.53; -0.22]) [15,18]. Matching the MyoGrip and MyoPinch protocol with that of a previously published natural history cohort allowed for comparison of change in grip and pinch strength over a six month period, yielding a statistically significant improvement on grip (p = 0.03) and pinch (p = 0.003) strength [19]. The lack of decline in these measures with ATL1102 during the trial period is once again encouraging and warrants further investigation.

### MRI of upper limb

Magnetic Resonance Imaging of muscle is increasingly used as a biomarker for disease stage and progression. The most widely used scoring method is the Mercuri Score, which requires a skilful investigator to visually score the chosen muscles based upon a standardised set of criteria encompassing degree of atrophy, oedematous changes and fatty infiltration of the muscle, to create an aggregate score. The more recent development of automated fat fraction analysis reduces the inter-user variability and provides a more quantitative measure of assessment. Matching the MRI protocol with that of a previously published natural history cohort allowed for direct comparison of change in fat fraction over a six month period [19]. From this published natural history data, disease is expected to progress with a mean increase of central forearm muscle fat fraction percentage of 3.9% (95%CI 1.9,5.7) over six months. The apparent trend towards a decrease in mean forearm muscle fat fraction of 0.52% (95% CI -5.62, 4.58; Median 1.4%) seen after six months of treatment with ATL1102 may suggest that ATL1102 could be modifying the rate of fatty infiltration into these muscles. These changes were replicated in the proximal and distal muscle groups. For future MRI studies it would be important to set a clear protocol with imaging tags placed over surface landmarks to ensure uniformity of subsequent scans.

### Conclusion

The proof of concept pre-clinical data supports a potential protective effect of an antisense oligonucleotide to CD49d RNA in the mdx mouse model of DMD. This phase 2 open-label clinical trial has shown that ATL1102 has a good safety profile and is well tolerated with minor injection site reactions the only treatment-related adverse events reported. The positive observations in functional efficacy outcomes suggesting stabilization, and results compared with historical natural history data, particularly the PUL2.0, MyoGrip, MyoPinch and MRI fat fraction analysis, justifies the ongoing drug development program of ATL1102 in non-ambulant boys with DMD and provides a rationale to proceed with larger placebo-controlled studies of this novel therapeutic agent.

### Supporting information

**S1 Checklist. TREND statement checklist.**
(PDF)

**S1 Fig. Inclusion and Exclusion criteria for the clinical trial.**
(DOCX)

**S2 Fig. Scatter Plot showing the association between the Moviplate scores and the PUL 2.0 distal dimension scores over the 24 Week ATL1102 treatment period with a linear regression line plotted.**
(TIF)

**S3 Fig. Scatter plot showing the association of the Grip Strength scores and the MRI data of the forearm lean muscle area (non-fat) over the 24 week ATL1102 treatment period: The MRI is the determined lean muscle mass compartment across the mid (central) dominant forearm.** The linear regression line is also plotted.
(TIF)

**S4 Fig. Gating strategy for identification of murine CD4+ and CD8+ T cell populations isolated from spleen of mdx mice treated with ISIS 348574.** A) Singlets (FSC-H vs FSC-A), B) Live cells (Propidium Iodide vs FSC-A) and C) Lymphocytes (SSC-A vs FSC-A) were gated to remove doublets, dead cells, debris and large/granular cells. D) Anti-CD4-V450 and anti-CD8a APC-Cy7 to were used to gate populations of CD4+ and CD8+ T cells.
(TIF)

**S5 Fig. Correlation of the Moviplate scores and the PUL 2.0 distal dimension scores over the 24 Week ATL1102 treatment period.**
(TIF)

**S6 Fig. Correlation of the Grip Strength scores and the MRI data of the lean muscle area (non-fat) across the mid (central) dominant forearm over the 24-week ATL1102 treatment period.**
(TIF)

**S7 Fig. Table of Participant specific genetic variant within Dystrophin Gene.**
(DOCX)

**S1 File.**
(PDF)

**S2 File.**
(PDF)

**S3 File.**
(PDF)

**S4 File.**
(PDF)

## Acknowledgments

The authors wish to acknowledge the contribution of Isabelle Ledoux and Simone Birnbaum of the Insitute of Myology, Paris, France who provided support with the quality control and analysis of the MyoSet functional outcome measures; Valeria Ricotti of the Dubowitz Neuromuscular Centre, London UK, who provided support on the comparative analysis of the MRI observations; and Annabell Leske and Vicky Beal and the team at Avance Clinical, Adelaide, Australia.

## Author Contributions

**Conceptualization:** George Tachas, Peter J. Houweling, Monique M. Ryan.

**Data curation:** Ian R. Woodcock, Peter J. Houweling, Michael Kean, Jaiman Emmanuel.

**Formal analysis:** Ian R. Woodcock, Nuket Desem, Jaiman Emmanuel, Chantal Coles, Chrystal Tiong, Peter Button, Jean-Yves Hogrel.

**Funding acquisition:** George Tachas.

**Methodology:** Ian R. Woodcock, George Tachas, Peter J. Houweling, Chrystal Tiong, Monique M. Ryan.

**Project administration:** Ian R. Woodcock, Nuket Desem, Peter J. Houweling, Rachel Kennedy, Kate Carroll, Katy de Valle, Sarah Catling-Seyffer.

**Resources:** George Tachas, Shireen R. Lamandé.

**Software:** Michael Kean, Peter Button.

**Supervision:** Peter J. Houweling, Kate Carroll, Shireen R. Lamandé, Daniella Villano, Monique M. Ryan, Martin B. Delatycki, Eppie M. Yiu.

**Writing – original draft:** Ian R. Woodcock, George Tachas, Peter J. Houweling, Eppie M. Yiu.

**Writing – review & editing:** Ian R. Woodcock, George Tachas, Nuket Desem, Peter J. Houweling, Michael Kean, Jaiman Emmanuel, Rachel Kennedy, Kate Carroll, Katy de Valle, Justine Adams, Shireen R. Lamandé, Chantal Coles, Chrystal Tiong, Matthew Burton, Daniella Villano, Peter Button, Jean-Yves Hogrel, Sarah Catling-Seyffer, Monique M. Ryan, Martin B. Delatycki, Eppie M. Yiu.

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
