## [Decision Letter · Decision Letter 0]

6 Sep 2023

PONE-D-23-13519A Phase 2 open-label study of the safety and efficacy of weekly dosing of ATL1102 in patients with non-ambulatory Duchenne muscular dystrophyPLOS ONE

Dear Dr. Woodcock,

Thank you for submitting your manuscript to PLOS ONE. After careful consideration, we feel that it has merit but does not fully meet PLOS ONE’s publication criteria as it currently stands. Therefore, we invite you to submit a revised version of the manuscript that addresses the points raised during the review process.

We look forward to receiving your revised manuscript.

Kind regards,

Julie Dumonceaux

Academic Editor

PLOS ONE

Journal Requirements:

2. Thank you for stating the following in the Competing Interests/Financial Disclosure * (delete as necessary) section:

"The ATL1102 in DMD clinical trial was funded in its entirety by the sponsor Antisense Therapeutics Ltd. Authors Ms Desem and Dr Tachas are employees of the sponsor and so received payment for services from the sponsor as employees. Ms. Desem and Dr Tachas hold an equity interest in the sponsor. Dr Tachas and Ms Desem along with other sponsor employees and sub-contracted specialists were involved in the study design and data analysis. Authors Dr Woodcock and Dr Ryan were at the time of the trial employees of the Royal Children’s Hospital and Murdoch Children’s Research Institute and are not affiliated with the sponsor in any way and have not received any direct personal payment or honoraria from the sponsors, nor do they or their family members hold a financial interest or stock in the sponsor company. Dr Woodcock is still an employee of the above institutions, but Dr Ryan has since left the employment to take up public office. Dr Woodcock and Dr Ryan were involved in the trial design as unpaid consultants. As this was a clinical trial, publication was always planned from trial inception. No employees of the sponsor were involved in the data collection, although Ms Desem did liaise closely with the MCRI/RCH site staff and Clinical Trial Organisation throughout the trial.

Author Dr Button was paid for services as the study statistician.

None of the other authors received any payment from the sponsor to conduct this study. All other authors had input into writing or revising this manuscript.  

We note that you received funding from a commercial source: "

Antisense Therapeutics Ltd"

"The authors confirm that the involvement of employees of the sponsor Antisense Therapeutics Ltd in the trial design, data analysis and decision to publish this data does not alter our adherence to PLOS ONE policies on sharing data and materials."

4. We note that the original protocol that you have uploaded as a Supporting Information file contains an institutional logo. As this logo is likely copyrighted, we ask that you please remove it from this file and upload an updated version upon resubmission.

Reviewers' comments:

Reviewer's Responses to Questions

**Comments to the Author**

1. Is the manuscript technically sound, and do the data support the conclusions?

Reviewer #1: Yes

Reviewer #2: Yes

2. Has the statistical analysis been performed appropriately and rigorously? 

Reviewer #1: I Don't Know

Reviewer #2: Yes

3. Have the authors made all data underlying the findings in their manuscript fully available?

Reviewer #1: Yes

Reviewer #2: Yes

4. Is the manuscript presented in an intelligible fashion and written in standard English?

Reviewer #1: Yes

Reviewer #2: Yes

5. Review Comments to the Author

Reviewer #1: Thank you for this interesting paper. I do accept this paper, although following the title I found it confusing to discover that two studies were presented where one included mice. Given the study design and small cohort I woud highlight this 2 front study in the title.

Reviewer #2: Important note: This review pertains only to ‘statistical aspects’ of the study and so ‘clinical aspects’ [like medical importance, relevance of the study, ‘clinical significance and implication(s)’ of the whole study, etc.] are to be evaluated [should be assessed] separately/independently. Further please note that any ‘statistical review’ is generally done under the assumption that (such) study specific methodological [as well as execution] issues are perfectly taken care of by the investigator(s). This review is not an exception to that and so does not cover clinical aspects {however, seldom comments are made only if those issues are intimately / scientifically related & intermingle with ‘statistical aspects’ of the study}. Agreed that ‘statistical methods’ are used as just tools here, however, they are vital part of methodology [and so should be given due importance]. I look at the manuscript in/with statistical view point, other reviewer(s) look(s) at it with different angle so that in totality the review is very comprehensive. However, there should be efforts from authors side to improve (may be by taking clues from reviewer’s comments). Therefore, please do not limit the revision only (with respect) to comments made here.

COMMENTS: Although this manuscript is well drafted [and the study is excellent with respect to most of the aspects], I have few very minor observations/concerns (different opinion) which are given below:

In section ‘Pre-clinical studies in mice’ (lines 112) it is stated that “Male mdx (n = 48) and age matched C57Bl10/J wild-type (WT, n = 12) controls” which I have not understood (clearly). According to my information, sample size of matched samples generally is same. If it is not a standard ‘pair matching’, what type of matching it is? If it is 4 x 1 [4cases, 1control] matching {which is very much valid}, why not make that clear? I think, even for ‘Pre-clinical studies’ the methodology should be rigorously followed.

Since Consort (line 138) is a short form of the term {group of words} [‘CONsolidated Standard Of Reporting Trials (CONSORT)], it should be represented in ‘CAPITAL’ letters, in my opinion.

Is the statement made in lines 196-7 “A sample size of 9 participants was calculated as required to achieve a power of 80%” is correct? Consideration of “power” in Phase-II trials is questionable / rarely seen, in my experience. Most of the known formulas and sample size estimation software are available for Phase-III trials, in my knowledge. Which software or formula is used need to be mentioned.

Except these minor points, the article is acceptable. However, mind you that as pointed out in ‘important note’ above “This review pertains only to ‘statistical aspects’ of the study and so ‘clinical aspects’ should be assessed separately/independently. ‘Minor Revision’ is recommended.

6. PLOS authors have the option to publish the peer review history of their article (what does this mean?). If published, this will include your full peer review and any attached files.

Reviewer #1: No

Reviewer #2: No

---

## [Author Response · Author response to Decision Letter 0]

5 Oct 2023

In response to reviewer 1 comments: title to be changed to: “A Phase 2 open-label study of the safety and efficacy of weekly dosing of ATL1102 in patients with non-ambulatory Duchenne muscular dystrophy and pharmacology in mdx mice”

In response to reviewer 2 comments: There were a total of 48 mdx mice and 12 controls purchased from JAX laboratories and these were then split into 4 groups of n = 12 mdx. Wording in the manuscript clarified. 

The manuscript has been amended to include the software used: “Using nQuery (Version 8.5.2.0, Table MOT1-1 Paired t-test for differences in means, Statistical Solutions Ltd.), a sample size of 9 participants was calculated as required to achieve a power of 80%”.

---

## [Editor Report · Decision Letter 1]

10 Nov 2023

A Phase 2 open-label study of the safety and efficacy of weekly dosing of ATL1102 in patients with non-ambulatory Duchenne muscular dystrophy

PONE-D-23-13519R1

Dear Dr. Woodcock,

We’re pleased to inform you that your manuscript has been judged scientifically suitable for publication and will be formally accepted for publication once it meets all outstanding technical requirements.

Kind regards,

Julie Dumonceaux

Academic Editor

PLOS ONE
---

## [Editor Report · Acceptance letter]

20 Nov 2023

PONE-D-23-13519R1 

A Phase 2 open-label study of the safety and efficacy of weekly dosing of ATL1102 in patients with non-ambulatory Duchenne muscular dystrophy and pharmacology in mdx mice 

Dear Dr. Woodcock:

I'm pleased to inform you that your manuscript has been deemed suitable for publication in PLOS ONE. Congratulations! Your manuscript is now with our production department. 

Kind regards, 

on behalf of

Dr. Julie Dumonceaux 

Academic Editor

PLOS ONE